# Examining the potential influence of crosslinguistic lexical similarity on word-choice transfer in L2 English

**Itamar Shatz**[1]*, **Theodora Alexopoulou**[1], **Akira Murakami**[2]

**1** Department of Theoretical and Applied Linguistics, University of Cambridge, Cambridge, United Kingdom,
**2** Department of English Language and Linguistics, University of Birmingham, Birmingham, United Kingdom

* is442@cam.ac.uk

**Data Availability Statement:** All the data and code are available in the following Open Science Framework (OSF) repository: https://doi.org/10. 17605/OSF.IO/5EUA8.

## Abstract

We examined whether and how L1-L2 crosslinguistic formal lexical similarity influences L2 word choice. Our sample included two learner subcorpora, containing 8,500 and 6,390 English texts, written in an educational setting, by speakers of diverse L1s in the A1–B2 CEFR range of L2 proficiency. We quantified similarity based on phonological overlap between L1 words and their L2 (English) translations. This similarity relates to psycholinguistic *cognancy*, which occurs when words and their translations share a high level of formal similarity, often due to historical cognancy from shared etymology or language contact. We then used mixed-effects statistical models to examine how this similarity influences the rate of use of the L2 words; essentially, we checked whether L2 words that are more similar to their L1 translations are used more often. We also controlled for potential confounds, including the baseline L1 frequency of the English words. The type of crosslinguistic similarity that we examined did *not* influence learners' choice of L2 words in their writing in the present sample, which represents a type of educational setting that many learners encounter. This suggests that the influence of such similarity is constrained, and that communicative needs can override transfer from learners' L1 to their L2, which raises questions regarding when and how else situational factors can influence transfer.

## Introduction

### Theoretical background

Learners' native language (L1) influences their knowledge of and engagement with second language (L2) vocabulary, in terms of operations such as recognition, interpretation, storage, and retrieval. This is often attributed to *lexical transfer* [1], a type of *language transfer* or *crosslinguistic influence* [2–4]. Transfer can be *positive* when it facilitates language acquisition or use, for example because an L2 linguistic structure (e.g., a certain word) is identical to a corresponding structure in a learner's L1, which makes it easier for the learner to use it. Transfer can also be *negative* when it hinders language acquisition or use, in which case it is sometimes

**Funding:** I.S. received financial support from Hughes Hall at the University of Cambridge (https://www.hughes.cam.ac.uk/) and Cambridge Assessment English (https://www.cambridgeassessment.org.uk/). T.A. received financial support from the Isaac Newton Trust at the University of Cambridge (https://www.newtontrust.cam.ac.uk/) and EF Education First (https://www.ef.co.uk/). The funders had no role in study design, data collection and analysis, decision to publish, or preparation of the manuscript.

**Competing interests:** The authors have declared that no competing interests exist.

called *interference*; this can occur, for example, because an L2 structure is very different from a corresponding structure in a learner's L1, which makes it harder for the learner to use it.

A notable aspect of lexical transfer is that crosslinguistic similarity in form—i.e., *formal similarity* in phonology and/or orthography—between L1 words and their L2 translations facilitates the processing, acquisition, and use of the L2 words [1, 2, 5–12]. This similarity is usually conceptualized based on the overlap in sounds and/or letters between words in different languages. For example, the French word for "orange" is also spelled "orange" (though pronounced slightly differently), so it has higher formal similarity with its English translation than does the French word for "lemon" ("citron"). Accordingly, it will generally be easier for French speakers to acquire the English word "orange" than the word "lemon".

When two words with similar meanings across languages have a high level of formal similarity, they can be considered to be psycholinguistic *cognates*, though there is no exact threshold for cognancy based on similarity. Psycholinguistic cognancy frequently occurs because the words are also historical cognates, meaning that they share a common etymology, though cognancy may also involve words that were borrowed during language contact [11, 13, 14].

The facilitative effect of formal crosslinguistic similarity—referred to as the *cognate facilitation effect* when it involves cognates—is well-attested in the psycholinguistic and second-language acquisition (SLA) literature, and has been attributed to various cognitive mechanisms. The general explanation for it is that similarity in form between L1 and L2 words that share similar meanings facilitates the linking and/or mapping of L2 words to their L1 counterparts or to shared concepts, which facilitates the transfer of linguistic (e.g., semantic, syntactic, and morphological) information from the L1 to the L2 [1, 7–10, 15, 16].

Like most types of crosslinguistic influence, this form of lexical transfer is expected to play a role primarily during early stages of SLA, when learners rely more on their L1 in order to form and use their mental lexicon. However, this influence can also play a role at advanced stages of SLA, and therefore affect even highly proficient L2 learners [1, 5, 6, 10, 15].

Since previous studies on this crosslinguistic influence focused on L2 processing (e.g., recognition, comprehension, and retrieval), it remains unclear whether and how this effect extends to L2 production, especially since various factors might play a different role in processing than in production. For example, the goal and context of communication might play a greater role in L2 production (e.g., word choice in essays) than in many experimental processing paradigms (e.g., reaction time to isolated words).

There is evidence that increased overall lexical similarity between languages improves learning outcomes, thus leading to higher scores in L2 proficiency tests [17–19]. This could be due to facilitated processing of L2 words, which can, in turn, facilitate general acquisition, since the more words learners understand, the more input they can decipher, and the more easily they acquire words and other structures [20]. However, this finding is based on similarity between languages as a whole, and on composite L2 proficiency scores that involve a mix of factors, including vocabulary and grammar. Accordingly, it does not tell us if similarity across L1-L2 words influences the production and choice of individual L2 words.

Some evidence regarding this comes from studies of *word choice transfer*, a type of lexical transfer whereby a person's knowledge of a language influences their choice of words in another language [21–24]. This transfer means that learners' use of specific words and phrases—referred to as *lexical signature*, *lexical style*, or *wordprints*—can be used in stylometry to aid L1 identification [21, 22]. This applies both to relatively constrained settings such as TOEFL essays (which we will call *task-based settings*), where communication is fairly limited in terms of factors like the permissible topic and style, as well as to more *spontaneous settings*, where the topic and style of communication are not as constrained (e.g., when people are allowed to talk about almost whatever they want). However, studies on word-choice transfer generally only

investigated whether learners' L1 influences their choice of L2 words, but did not investigate what factors specifically drive this crosslinguistic influence.

One exception is Rabinovich et al. (2018), who showed that L1-L2 similarity can influence L2 word choice [13]. Specifically, they investigated the relatively spontaneous productions on a social media website (Reddit) of highly proficient (near-native) L2 English speakers of various Indo-European L1s. They focused on English words that were part of a *synset*, which is a set of multiple synonyms that correspond to the same meaning. Specifically, Rabinovich et al. focused on synsets where the synonyms had at least two different etymological paths (under the assumption that etymological cognancy generally leads to increased formal similarity), and the synonyms themselves were fairly interchangeable. They found clear evidence of a cognate facilitation effect, meaning that the speakers were more likely to use English words that are cognate with their L1 translation. For more information on this study, see Appendix S3 in S1 File (under "Analysis of synonym sets").

However, there is also evidence suggesting that the effect of crosslinguistic similarity might not extend to productions in task-based settings. Specifically, Crossley and McNamara (2011) found that L2 texts written by speakers with different L1s had similar scores on several global lexical measures, such as lexical diversity and polysemy, despite different levels of similarity between their L1s and the target L2 [25]. This is based on 599 L2 English texts in the *International Corpus of Learner English* (ICLE), written by Czech, Finnish, German, and Spanish speakers, who are "high intermediate to advanced" L2 English speakers (p. 274), and who wrote the texts as a response to one of few prompts for argumentative essays. A potential explanation for this finding is that, in these relatively constrained task-based settings, learners choose to use only words that are sufficiently relevant for their communication, regardless of which words are easier to use due to crosslinguistic similarity. Essentially, the factors constraining the communication (e.g., narrow communicative goals or the necessary formality level) may serve as situational and contextual factors that override the transfer from learners' L1 [21]. But, it is unclear if this is indeed the case, or if the findings of Crossley and McNamara can be attributed to a different factor, such as that they focused on global lexical measures, rather than on the use of individual words.

To summarize, there is clear evidence of a facilitative effect of L1-L2 lexical similarity on L2 processing, comprehension, and learning, particularly at the early stages of SLA [2, 17], and there is also evidence that learners' L1 can influence their L2 word choice [13, 22]. However, evidence regarding the influence of crosslinguistic *similarity* on L2 word choice is limited and less clear, especially in task-based settings.

## Research questions

We investigate the potential influence of crosslinguistic formal lexical similarity on word choice in a task-based English-as-a-foreign language (EFL) educational setting, to answer the following questions:

1. Does increased similarity in form between L1 words and their L2 translations lead to increased use of the L2 words in this setting?

2. If there is an effect of crosslinguistic similarity in such task-based settings, is it moderated by learners' L2 proficiency?

Answering these questions will help determine whether the effect identified by Rabinovich et al. [13] extends to task-based settings (we compare our approach with theirs in Appendix S3 in S1 File—"Comparison of our approach with that Rabinovich et al."). Furthermore, it will help determine whether findings regarding word-choice transfer in task-based settings are

likely attributable to some degree to crosslinguistic similarity, and whether the lack of L1 effect (i.e., *intergroup homogeneity*) found by Crossley and McNamara [25] is simply a feature of the global lexical measure that they used and/or their sample. In addition, the focus on an educational EFL setting will shed light on the influence of crosslinguistic similarity in this type of common environment, where, as we will see, there are often strong task effects on word choices.

## Our approach

We examined how formal similarity between L2 English words and their L1 translations influences the usage rates of the L2 words. For example, we wanted to see if, in a task dealing with food, an Italian learner of English will be more likely to use the word "lemon" than a French speaker, because the Italian word for "lemon" ("limone") sounds more similar to the English word than the French one ("citron") does. If similarity plays a role in this context, then we expected that learners—especially beginners—will prefer using similar words, because they are easier for them to process.

To investigate this, we constructed lists of L1-L2 word pairs, containing words in various L1s (e.g., German and French), together with their corresponding translations in English as the target L2 (e.g., citron-lemon). Then, we calculated the formal similarity between the words in each L1-L2 pair, based on the phonological overlap of the sounds that the words contain, where increased overlap denotes increased similarity (i.e., decreased lexical distance). We also found the baseline frequency of the target English words, to control for it in our analyses.

Next, we took a large-scale EFL learner corpus, containing texts written in response to various writing tasks, by learners with diverse L1s and varied L2 proficiency. Using the L1-L2 wordlists from the previous stage, we counted the number of times each target English word from the wordlists appeared in each text.

Finally, we built mixed-effects statistical models, to determine whether the rate of use of the target English words in the texts was predicted by the lexical similarity between each English word and its L1 translation, and whether this effect was moderated by L2 proficiency (to check if the effect of similarity is stronger at lower L2 proficiency levels). Our models controlled for relevant background variables, including the baseline frequency of the English words, as well as *task* and *item* effects. Ultimately, our key question was whether, all things being equal, L2 words that are more similar to their L1 translations will be used at higher rates, compared to words that are less similar.

## Methodology

Data and code are available at the following *Open Science Framework* (OSF) repository: https://doi.org/10.17605/OSF.IO/5EUA8

### Crosslinguistic similarity/distance

**Distance datasets.** We quantify crosslinguistic formal similarity based on the phonological distance between L1 words and their L2 translations, where increased distance denotes lower similarity. We will henceforth refer to this as *lexical distance*, though we use phonological distance as a proxy of overall lexical distance, which subsumes other types of similarity; for more information on this choice of terminology, see Appendix S1 in S1 File (under 'The term "lexical distance"'). To do this, we use two datasets, which contain lists of corresponding words in different languages, as outlined briefly below. For more information on these datasets and their processing, see the "Lexical-distance datasets information" document in the study's OSF repository.

The first lexical-distance dataset is the *Automated Similarity Judgment Program* (ASJP) [26]. It contains Swadesh lists, which are often used by researchers to calculate the lexical distance between languages [e.g., 18], and which contain words representing various concepts, such as *hear*, *water*, *full*, *one*, and *dog* [26, 27].

The Swadesh lists in the ASJP focus on a subset of 40 concepts; to control for variation in the completeness of the Swadesh lists across languages, we included in our analysis only the 38 concepts that are shared by all the languages in our sample. These languages, which are based on the ones available in the learner sample that is outlined later, are: Arabic, French, German, Italian, Japanese, Mandarin, Portuguese, Russian, and Spanish as L1s, and English as the target L2. In addition, we focus on single-word entries, in line with most prior research and to avoid potential confounds, and so we included only entries that do *not* contain a multi-word phrase in any of the L1s or English. Accordingly, the final Swadesh-based sample contains 225 entries, with 25 entries for each of the 9 L1s, where each entry is a row containing an English word together with all its L1 counterparts in a specific L1.

The second lexical-distance dataset that we use is the *Intercontinental Dictionary Series* (IDS), which contains parallel dictionaries in various languages [28]. Similarly to the Swadesh lists, this dataset also contains a standardized list of words and their corresponding counterparts in various languages. The parallel dictionaries contain substantially more words per language than the Swadesh lists (~1,300 general word meanings compared to ~40). However, they contain data only for French, German, Italian, Portuguese, and Spanish (out of the L1s in the present sample). Accordingly, they complement the Swadesh lists, but do not replace them.

As with the Swadesh lists, we included only single-word entries in our analysis of this dataset. Furthermore, we removed from the parallel dictionaries a small number of words (22) that also appeared in the Swadesh lists, so that the words in each dataset were unique. Accordingly, the final parallel-dictionaries sample contains 5,515 entries, with 1,103 entries for each of the 5 L1s, where each entry is a row containing an English word and all its L1 counterparts in a single L1.

Our approach aligns in this regard with Rabinovich et al. (2018), who created their wordlist (with 1,143 words) based on their lexical dataset (*Etymological WordNet*) rather than their learner sample, though they did use learner data when choosing the most prominent sense of a word in cases where multiple parts-of-speech categories were available. An alternative potential approach for creating these wordlists is to base them on the words that appear in our learner sample. However, this could bias the analyses, since the presence and absence of words from the sample can be an important signal regarding associated crosslinguistic influence.

To illustrate this, consider a simple situation, where we compare, among German learners, the rate of use of two English words, with an equally low baseline frequency. One of the English words is similar to its German translation, whereas the other one is dissimilar. If there is indeed a facilitative effect of similarity, then we might expect that the dissimilar word will not be used by learners (because it has low baseline frequency), but that the similar word will be used despite the low baseline frequency (because of the facilitative effect). However, if we remove the distant word from our analysis because it was not used at all, then we would be obscuring the effects of similarity by comparison. Essentially, the fact that a word is not used at all by learners is important to our analyses, as it allows us to more accurately assess the effects of distance.

**Calculating lexical distance.** The lexical-distance measure that we use is *Levenshtein distance normalized* (LDN). Extensive information about this measure, including its psycholinguistic validation, is presented in Appendix S1 in S1 File (under "Validation of Levenshtein distance"), and is also summarized below.

Intuitively, LDN generally represents the degree of phonological or orthographic overlap between two words. It is calculated by taking the minimum number of character substitutions,

additions, and deletions that are needed to transform one string to another (i.e., the *Levenshtein distance*), and dividing it by the length of the longer string, to account for variations in word length. For example, in the case of the word *knee*, the English-German pair /ni/-/kni/ has an LDN of 0.33, since there is 1 character transformation (a /k/ is inserted or deleted), and the length of the longer string is 3. By contrast, the LDN for the corresponding English-Japanese pair /ni/-/hiza/ is greater (0.75), since there is less overlap, so more transformations are needed.

In the present research, we first calculated lexical distance between each L1 entry and its corresponding L2 (English) entry, based on their phonological (IPA) transcription. When there were multiple L1 synonyms available (e.g., "soil" in French—*sol* and *terre*), we used the distance from the closest synonym, as our goal was to identify cases where the L2 word is closely similar to an L1 word (and is likely also cognate with it).

We used phonological—rather than orthographic—overlap as a measure of distance, because this enables us to examine distance from L1s that have a substantially different script from English, like Arabic and Mandarin. Nevertheless, in the parallel-dictionaries sample, where all the L1s share English's Latin script, there was a strong correlation between phonological and orthographic overlap ($r = .68$, *95% CI* = [.67, .70], $p < .001$). This aligns with findings of other research [11, 29, 30], like an $r = .782$ found in a dataset of English and Spanish words [11]. This strong correlation suggests that the phonological overlap that we found for L1s that share English's script is indicative of the associated orthographic overlap for these L1s, so even if a large part of the effect of similarity is due to overlap in orthography, we would expect to detect it in our analyses.

We used LDN for several reasons. First, it can be calculated in an automated, objective, and replicable manner for a large number of words from different languages [14]. Second, it is the most conventional measure that is used for this purpose, and, as shown in detail in Appendix S1 in S1 File (under "Validation of Levenshtein distance"), it has been extensively validated, including through correlations with other measures of language distance, such as expert cognancy judgments from historical linguistics and perceived language distance from psycholinguistics [14, 31]. Furthermore, LDN was used by other SLA researchers [e.g., 30] to quantify crosslinguistic similarity between individual words—often to distinguish cognates from noncognates when investigating cognate facilitation at the word level. It was also found to be a robust predictor of relevant L2 outcomes, including word recognition [11] and retrieval [32].

However, LDN also has some important limitations, discussed in Appendix S1 in S1 File (under "Limitations of LDN"), which we will also briefly outline below.

The first limitation is that LDN treats all character transformations as equal, even though some transformations are less "substantial" phonologically than others. We partially addressed this by replicating our analyses using *feature edit distance* (Appendix S2 in S1 File).

The second limitation is that our use of LDN only considers one aspect of formal similarity (phonological overlap), but other formal factors (e.g., orthographic depth) and non-formal factors (e.g., semantic/pragmatic similarity), may also affect crosslinguistic influence. Nevertheless, past studies [e.g., 32] found a facilitative effect of formal similarity even without considering such factors, as did Rabinovich et al. [13], who did not investigate the influence of these factors. Furthermore, we used mixed-models to control for some of these potential effects, and replicated our analyses on a sub-sample containing only German speakers ("German-only models" in Appendix S5 in S1 File), to minimize the influence of some of these factors (e.g., variation in the effects of similarity across language families).

Finally, LDN does not assess cognancy directly, which we use in the psycholinguistic sense, of words that have similar meaning and pronunciation/spelling across languages. Rather, LDN only quantifies the formal similarity between words that are generally similar in terms of

meaning. Nevertheless, as noted above, LDN is strongly correlated with cognancy, and has been used to estimate cognancy in SLA studies that then used it to successfully predict L2 outcomes, including at the word level [11, 32], so we expect to be a reasonable approximation in the context of the present large-scale analyses.

These limitations are important to keep in mind. However, given the ways we addressed them (as outlined above and in "Limitations of LDN" in Appendix S1 in S1 File), and given the validation for the use of LDN in the manner we are using it (as outlined above and in "Validation of Levenshtein distance" in Appendix S1 in S1 File), we believe that the use of LDN is reasonable in the present study. Notably, even if it will be unable to perfectly capture *all* of the effects of crosslinguistic similarity, it should be able to successfully capture *some* of them, as it did in many past SLA studies.

Likewise, although we did not focus on the effects of similarity on word choice within synonym sets in particular (unlike Rabinovich et al.), our sample does include such sets, as shown in Appendix S3 in S1 File (under "Analysis of synonym sets"). Given this, we would expect to find at least some effect of crosslinguistic similarity in the sample, even if it is confined only to such sets, though we do not claim that this is necessarily the case.

**Lexical distances.**   Fig 1 and Table 1 contain information about the the lexical distances between the L1s in the sample and English. The distances of all word pairs are available in the data files in the OSF repository (under "Lexical distance & frequency data").

This figure and table show that the words in the datasets cover the full range of distances from English (0–1). However, most words are highly dissimilar (with an LDN at or near 1), even in L1s that are relatively lexically similar to English (e.g., German and French). This is important, since it suggests that in naturalistic settings, L2 learners may have limited opportunities to benefit from facilitative effects of crosslinguistic lexical similarity, so they must adapt to using L2 words that are dissimilar from their L1 translations.

Despite using this type of representative sample, there was a sufficient range of distances in our sample that the estimates of its effects were precise in our models, as shown in the results. Nevertheless, due to concerns that the high degree of dissimilarity might obscure the effects of crosslinguistic influence, we replicated our analyses using data from just German speakers, as this was the L1 that was the closest to English, and had the broadest range of LDN values. This analysis (presented in Appendix S5 in S1 File, under "German-only models"), is similar to the analyses of other researchers who analyzed L2 acquisition among speakers of a single L1, and in particular an L1 that is relatively similar to the L2, like De Wilde et al., who looked at the acquisition of L2 English among L1 Dutch speakers [30].

In addition, the distances are largely aligned with those based on general language classification. Specifically, the Germanic and Romance L1s are the closest to English, and the Indo-European L1s are closer to English than the non-Indo-European L1s, except that Japanese is shown as being closer to English than Russian is (so Japanese is closer than we would expect, and Russian is further than we would expect). However, because the lexical-distance datasets were modified through the removal of multi-word entries, the overall similarity between each L1 and English that is shown in this figure and table should *not* be interpreted as the mean similarity between that L1 and English. Indeed, as shown in Appendix S1 in S1 File (under "Validation of Levenshtein distance"), when the unmodified wordlists are used, meaning that multi-word entries remain in the sample, Japanese and Russian switch positions as expected, and consequently, all the Indo-European L1s are closer to English than the non-Indo-European L1s. Nevertheless, this is not important for our analyses, since we focus on the similarity and use of individual words, rather than on similarity at the language level and on global measures of word use (e.g., lexical diversity).

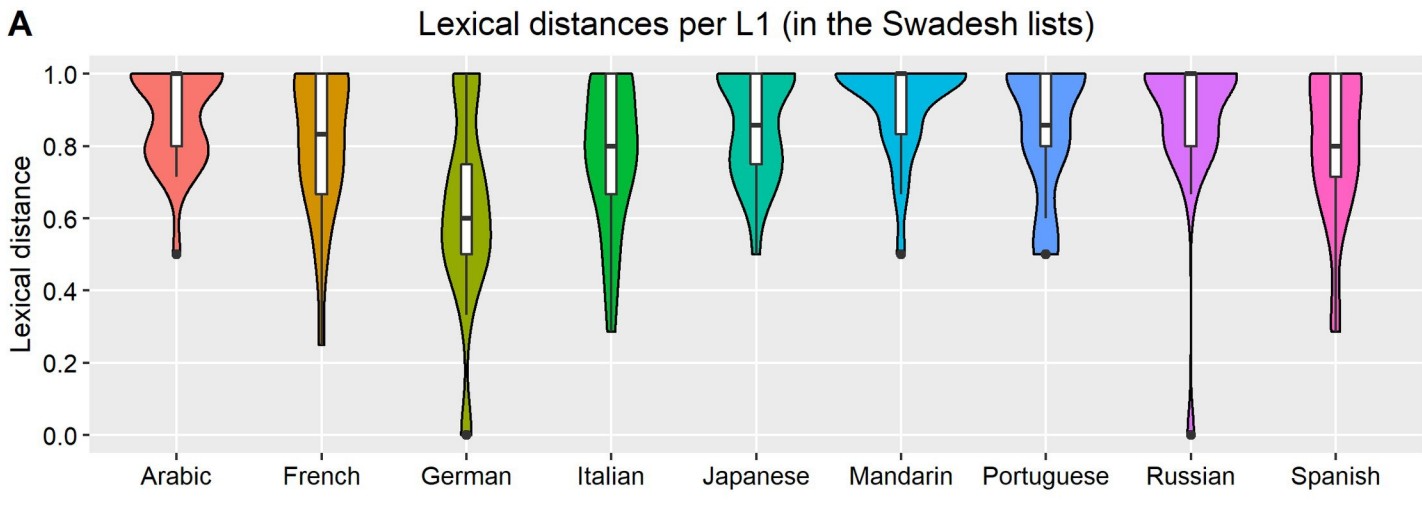

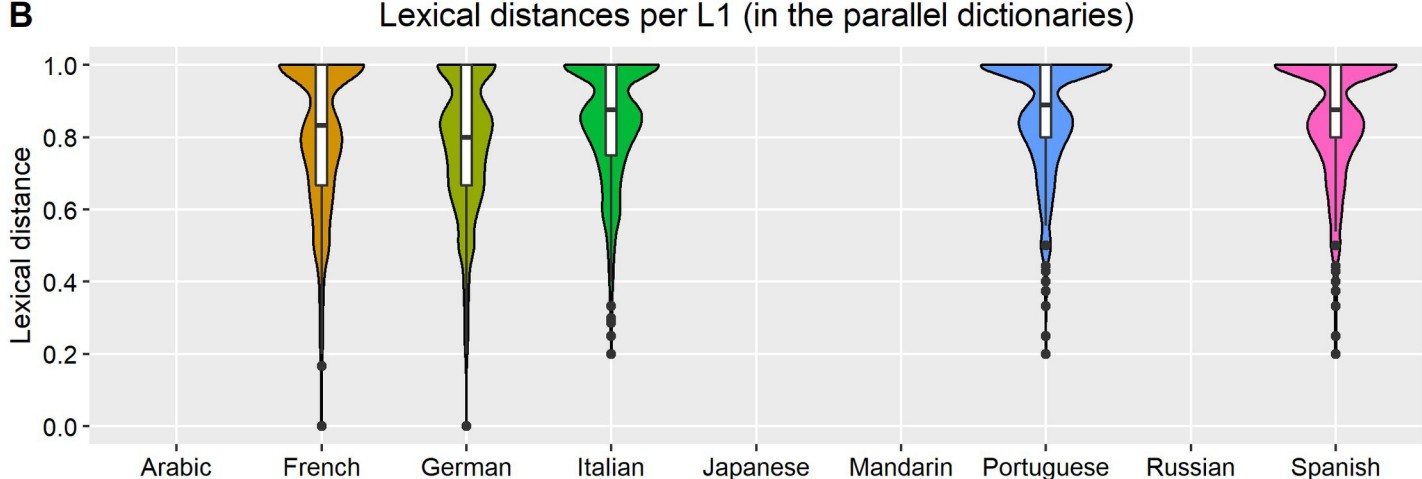

**Fig 1. Lexical distance between L1 words and English, per L1 in each dataset.** The distance is equal to the phonological LDN between L1 words and their most lexically similar English counterpart. Within the boxplots, the middle line indicates the median, the lower/upper hinges indicate the 1st/3rd quartiles, the whiskers indicate 1.5 interquartile ranges (IQR) past the hinges, and the dots indicate outliers. The violin plots indicate an estimate of the probability density of lexical distance for each L1, which can be viewed as the likelihood that a word in each L1 will have a certain lexical distance, where increased width indicates greater likelihood. Data is based on 25 words per L1 in the Swadesh lists and 1,103 words per L1 in the parallel dictionaries (after the removal of multi-word entries). These L1s were chosen based on the ones available in the learner sample, which is presented later.

## Baseline word frequency

*Baseline word frequency* represents how often an English word is used in general English. We need to control for this, since it can influence our response variable (the usage rate of L2 words). The "Baseline frequency information" document in the OSF repository contains detailed information about how we calculated this frequency. To summarize, we used the *wordfreq* library in Python [33], which curates frequency information from a number of diverse and large-scale sources, including books, subtitles, news, and social media. We used their *Zipf frequency* measure, developed by van Heuven et al. [34], which is the base-10 logarithm of the number of times a word appears per billion words (e.g., a Zipf value of 6 means a word appears once per thousand words).

Fig 2 shows the frequency distribution of the English words in our lexical-distance datasets. All frequencies are available in the OSF repository (under "Lexical distance & frequency data").

**Table 1. Statistics about the lexical distances between the L1s and English in each dataset.** L1s are arranged in order of increasing mean lexical distance in the Swadesh lists.

| L1 | Swadesh lists | | | | | Parallel dictionaries | | | | |
|---|---|---|---|---|---|---|---|---|---|---|
| | mean | SD | median | IQR | range | mean | SD | median | IQR | range |
| German | .622 | .27 | 0.60 | 0.50–0.75 | 0.00–1.00 | .785 | .18 | 0.80 | 0.67–1.00 | 0.00–1.00 |
| Italian | .776 | .20 | 0.80 | 0.67–1.00 | 0.29–1.00 | .847 | .16 | 0.88 | 0.75–1.00 | 0.20–1.00 |
| Spanish | .808 | .21 | 0.80 | 0.71–1.00 | 0.29–1.00 | .860 | .16 | 0.88 | 0.80–1.00 | 0.20–1.00 |
| French | .813 | .20 | 0.83 | 0.67–1.00 | 0.25–1.00 | .814 | .20 | 0.83 | 0.67–1.00 | 0.00–1.00 |
| Portuguese | .848 | .18 | 0.86 | 0.80–1.00 | 0.50–1.00 | .873 | .15 | 0.89 | 0.80–1.00 | 0.20–1.00 |
| Japanese | .864 | .15 | 0.86 | 0.75–1.00 | 0.50–1.00 | - | - | - | - | - |
| Russian | .881 | .21 | 1.00 | 0.80–1.00 | 0.00–1.00 | - | - | - | - | - |
| Arabic | .887 | .14 | 1.00 | 0.80–1.00 | 0.50–1.00 | - | - | - | - | - |
| Mandarin | .924 | .13 | 1.00 | 0.83–1.00 | 0.50–1.00 | - | - | - | - | - |

*Note.* The distance measure is based on the phonological LDN from the closest synonym, calculated for the single-word entries in each dataset. There were 225 entries in the Swadesh lists (i.e., rows with an English word and all its corresponding counterparts in a certain L1), with 25 entries for each of the 9 L1s in the dataset. There were 5,515 entries in the parallel dictionaries, with 1,103 for each of the 5 L1s. All counts are after the removal of multi-word entries.

The mean Zipf frequency in the Swadesh lists was 5.24 (SD = 0.72, median = 5.14, range = 4.15–7.11), and the mean Zipf frequency in the parallel dictionaries was 4.35 (SD = 0.83,

**Fig 2. The baseline (Zipf) frequency of the English words in each lexical-distance dataset.** Within the boxplots, the line inside the box indicates the median, the lower/upper hinges indicate the 1st/3rd quartiles, the whiskers indicate 1.5 IQRs past the hinges, and the dots indicate outliers. The violin plots indicate an estimate of the probability density of the frequency of English words. Data is based on 25 English words in the Swadesh lists and 1,103 words in the parallel dictionaries.

median = 4.32, range = 1.87–7.41). Accordingly, both datasets included a wide range of words with different frequencies, though this range was greater in the parallel dictionaries.

One concern regarding word frequencies was there will not be enough high-level (i.e., low-frequency) vocabulary words in the lists, which could be a problem if the effects of similarity are stronger in—or restricted to—low-frequency words. However, as shown in Appendix S3 in S1 File (under "Comparison of baseline word frequencies"), the distribution of the Zipf frequencies in our parallel-dictionaries sample—based on the mean, SD, and range—is similar to that of other studies that found a cognate facilitation effect, and our sample also contains substantially more (1,103) words, so this should not be an issue for our analyses. Furthermore, as explained in the "Data analysis" section of the paper and the "Added-interactions models" section in Appendix S5 in S1 File, we built supplementary models, which showed that there is no interaction between distance and frequency in our sample.

In addition, note that past studies found a cognate facilitation effect even when controlling for frequency [e.g., 8, 11, 30], as shown under "Correlations of distance, frequency, and word use" in Appendix S3 in S1 File. Accordingly, this effect does not appear to be simply the result of a frequency confound, and we would expect to find a similar effect in the present sample, even when controlling for frequency.

## Learner sample

In this section, we briefly outline the learner sample that we used. For more details on it, see the "Sample information" document in the study's OSF repository.

The learner sample came from the *EF-Cambridge Open Language Database* (EFCAMDAT), an open-access EFL learner corpus, containing texts written by learners in *Englishtown*—EF's online English school [35–37]. When a learner joins Englishtown, their English proficiency is determined through a dedicated placement test [35]. Based on this, they are placed at a starting proficiency level, and the EFCAMDAT spans 16 such levels, which EF has aligned with common proficiency standards [35], such as the Common European Framework of Reference for Languages (CEFR) [38]. Each level consists of several distinct lessons. After completing a lesson, learners are assigned a writing task that they submit online, and receive feedback on from a teacher. These tasks, which are described in more detail in the "Sample information" document (under "Background information on the EFCAMDAT"), cover a wide range of styles and topics, such as describing your favourite day, reviewing a song, writing an online profile, or giving instructions to a house-sitter. The curriculum is standardized, so learners with different L1s follow the same lessons and activities, and are given the same writing tasks. Note that we use the term "task" here in the sense in which it is generally used in the EFCAMDAT; as shown in the later explanation of our analyses, we do not make a claim regarding the influence of different specific aspects of the tasks, such as their genre [39].

For our analyses, we used the *EFCAMDAT Cleaned Subcorpus* [40]. The key feature of this dataset is that it is split into two subcorpora, each of which contains texts written by similar learners in response to different lessons and prompts. This means, for example, that both the first and the second subcorpora contain texts written by Mandarin learners in task #5, but the learners in the first subcorpus wrote their texts after a different lesson and in response to a different prompt than the learners in the second subcorpus. Accordingly, using this dataset presents two important advantages for research. First, it allows us to accurately categorize texts based on the task that they correspond to. Second, this offers an opportunity to analyze two similar but distinct learner samples, which serves as a form of replication.

We selected random texts from this dataset, in a balanced manner across L1s, proficiency levels, and tasks. For a full explanation of this process, see the relevant document in the OSF

repository (under "Sample selection process and final sample"). The final samples are outlined in Table 2.

## Word usage

To assess learners' use of L2 vocabulary, we calculated the number of times each English word in the lexical-distance datasets appears in any given text in the learner sample. We did this separately for each cross of one of the lexical-distance datasets with one of the EFCAMDAT subcorpora, as shown in Table 3. Note that we calculated counts based on a spelling-corrected version of each text, as discussed in Appendix S4 in S1 File (under "Spelling correction").

Statistics about the counts of target words appear in Table 4. For more information on the raw response variable, see the section on "Correlations of distance, frequency, and word use" in Appendix S3 in S1 File. In addition, we also built models looking only at the presence/absence of target words, as shown in Appendix S5 in S1 File (under "Binary-response models"), which replicated the results of the count-based models.

Broadly, the data can be characterized as having (1) a high proportion of zeros and (2) a right skew, which means that there were many cases where a target was not used in a text, and a small number of cases where a target word was used in a text multiple times. This means that most words are not used in most texts, and that some words are also not used in any of the texts, which is expected, given that we include specialized "high level" (i.e., low frequency) words in our sample. However, the inclusion of such words does not pose an issue for our models, as indicated by the model diagnostics that we discuss later, as well as the precise coefficient estimates for our predictors. In addition, note that removing such words from our sample would bias the results.

**Table 2. Final learner samples (with English as the target L2).**

| | |
|---|---|
| L1s [a] | Arabic, Japanese, Mandarin, Russian (these appear only in the Swadesh sample)<br>French, German, Italian, Portuguese, Spanish (these appear in both the Swadesh and parallel samples) [b] |
| L2 proficiency levels | EFCAMDAT 1–12 (equivalent to CEFR A1–B2) |
| Number of tasks per subcorpus | 95 (first) / 71 (second) [c] |
| Number of texts per L1 per task | 10 [d] |
| Number of texts per subcorpus | In Swadesh sample: 8,500 (first) / 6,390 (second)<br>In parallel sample: 4,747 (first) / 3,550 (second) [e] |

[a] L1s in the EFCAMDAT are estimated based on learners' nationality, an approach that has been used in previous studies and validated empirically, as shown in the OSF "Sample Information" document under "Background information on the EFCAMDAT".

[b] The nationality for Arabic is Saudi Arabian; for Mandarin—Chinese; for Portuguese—Brazilian; for Spanish—Mexican. For other L1s, the L1 is based on the corresponding nationality (e.g., Japanese).

[c] There are 8 tasks per EFCAMDAT level in the first subcorpus and 6 tasks per level in the second (with one task per lesson). An exception is task #51, in which texts from both subcorpora were placed in the first subcorpus due to the software used to classify them, so this task was removed from this sample.

[d] There were a few exceptions to this in the first subcorpus, which had 2–9 texts (mean = 6.43, SD = 1.79); these cases (14 out of 855, 1.64%) are listed in the OSF "Sample information" under "Cases with fewer than 10 text".

[e] The difference in the number of texts is because the parallel sample contains data for 5 L1s out of the original 9, and so contains 55.85% of the total texts available in the first subcorpus, and 55.56% of those available in the second subcorpus.

**Table 3. The four final samples, each representing a cross between a lexical-distance dataset and a subcorpus.** *Observations* equal the number of *words per L1* in a lexical-distance dataset times the number of *texts* available in the subcorpus.

| Distance dataset | Subcorpus | L1s | Words per L1 | Texts [a] | Observations |
|---|---|---|---|---|---|
| Swadesh lists | first | 9 | 25 | 8,500 | 212,500 |
| Swadesh lists | second | 9 | 25 | 6,390 | 159,750 |
| Parallel dictionaries | first | 5 | 1,103 | 4,747 | 5,235,941 |
| Parallel dictionaries | second | 5 | 1,103 | 3,550 | 3,915,650 |

[a] The number of texts available for the parallel-dictionaries samples reflects them containing data for 5 out of 9 L1s that we examine.

This distribution is common for count data, and is expected given the diverse range of tasks and words in our sample, including the spectrum of low- and high-frequency words. It should *not* be interpreted as indicating overdispersion or zero-inflation, since those are features of a model rather than the response variable [41]. Indeed, the assumption checking (in the "Model diagnostics" section of Appendix S4 in S1 File) show that the models are not overdispersed or zero-inflated; rather, some actually have underdispersion, though as shown in the aforementioned section, this does not substantially influence our results. Also, as noted in the next section, we used Poisson models in our analyses, since they are designed for dealing with this type of count data, and due to the large size of the samples, there was a sufficient number of "positive" observations (i.e., with a count $> 0$) that the models were able to converge properly.

Furthermore, our results—as well as the use of Poisson models—were supported by the supplementary logistic-regression models that we built, which used a binary response variable (as shown in Appendix S5 in S1 File, under "Binary-response models").

## Data analysis

We built *generalized linear mixed-models* (GLMMs), separately for each combination of subcorpus and lexical-distance dataset (e.g., Swadesh lists and the first subcorpus). Specifically, we built *Poisson* models (with the canonical *log* link), due to the use of count data in the response variable [42, 43]. The structure of the models was as follows (the formula we used appears in Appendix S4 in S1 File, under "Model formula"):

1. **Response variable:** *Rate of usage* of the target English word. This is based on the count of the target English word in a text (i.e., the number of times it appears in it), which is then *offset* by the total number of words in the text (specifically, it is offset by the *log* of the word-count—an exposure variable that is based on the *wordcount* variable in the EFCAMDAT

**Table 4. Statistics about the distribution of the count data that was used in the models (i.e., the number of times a word appeared in a text).** The specific statistics are given either for *total* cases, or for cases where the count was greater than zero (*count>0*).

| Dataset | Subcorpus | $N_{(total)}$ | $N_{(count>0)}$ | Prop.$_{(count>0)}$ [a] | Mean$_{(total)}$ | SD$_{(total)}$ | Mean$_{(count>0)}$ | SD$_{(count>0)}$ | Max |
|---|---|---|---|---|---|---|---|---|---|
| Swadesh | first | 212,500 | 13,049 | 0.061 | 0.174 | 0.968 | 2.832 | 2.782 | 24 |
| Swadesh | second | 159,750 | 9,819 | 0.061 | 0.188 | 1.104 | 3.063 | 3.323 | 26 |
| Parallel | first | 5,235,941 | 59,566 | 0.011 | 0.016 | 0.183 | 1.417 | 0.973 | 19 |
| Parallel | second | 3,915,650 | 47,072 | 0.012 | 0.017 | 0.196 | 1.452 | 1.058 | 15 |

*Note*. The difference in distributions between the parallel dictionaries and Swadesh lists could be attributed, at least in part, to the parallel dictionaries containing some lower-frequency words. Specifically, the mean Zipf frequency in the Swadesh lists was 5.24 (SD = 0.72, median = 5.14, range = 4.15–7.11), while the mean Zipf frequency in the parallel dictionaries was 4.37 (SD = 0.84, median = 4.35, range = 1.87–7.41).

[a] This represents the proportion of entries with a count greater than 0, out of all entries in the sample.

Cleaned Subcorpus—since the log is the canonical link function for Poisson models). This is needed to control for different texts having a different total number of words, and produces a rate at which target words occur per word in the text [42, 43]. In addition, we built supplementary models with a *binary* response variable, based on whether a target word was used in a text or not. Essentially, while the main models focused on the target words as *tokens*, by examining their counts, these models focused on them as *types*, by examining their presence/absence. These models replicated the results of the main models, as shown in Appendix S5 in S1 File (under "Binary-response models").

2. **Predictors**:

   a. *Lexical distance* (of individual L1-L2 word pairs), based on the phonological LDN between the English word and its closest synonym in the L1 of the learner who wrote the text.

   b. *L2 proficiency*, based on EFCAMDAT proficiency level (1–12, corresponding to CEFR A1–B2) of the learner at the time they wrote the text, as each task in the dataset is classified under a specific proficiency level. This predictor is used to statistically control for the inclusion of multiple L2 proficiency levels in the sample, and enables us to isolate the effects of lexical distance on the rate of use of the target rate, once L2 proficiency is accounted for. Essentially, it allows us to determine whether learners at the same L2 proficiency levels differ in their word choice, while including a range of L2 proficiency levels in our sample.

   c. *Interaction between lexical distance and L2 proficiency*, to see whether the effects of L2 proficiency moderate those of lexical distance, and especially whether lexical distance has a stronger effect at lower proficiency levels.

   d. *Word frequency* of each English word (based on its baseline frequency in the English language), to control for this factor when considering the word's rate of usage in the L2 texts. We also built supplementary models with potential interactions between *distance/ frequency*, *proficiency/frequency*, and *distance/proficiency/frequency*, which replicated the findings of the main models, as shown under "Added-interactions models" in Appendix S5 in S1 File.

3. **Random effects** (random intercepts unless noted otherwise):

   a. *Learner*, to control for learners who had more than one text in the sample. Most learners only had a single text in the sample (the mean number of texts per learner was 1.36 in the first subcorpus and 1.41 in the second). Multiple texts per learner were included to achieve sufficient coverage of the sample, in line with prior studies on the EFCAMDAT [e.g., 39, 44, 45]. See the "Sample information" document in the OSF repository for more details (under "Number of texts per learner").

   b. *L1*, with random slopes for *lexical distance*, to control for any additional effects from the learners' L1 and their associated (e.g., cultural) background.

   c. *Task*, to control for all the aspects of each writing task that can influence word choice, such as its prompt, with the exception of the task's associated L2 proficiency level, which we control for using the relevant predictor. This approach accounts for all aspects of task effects in aggregate, without disentangling its different aspects; for more information, see Appendix S4 in S1 File (under "Task random effect").

 d. *Word*, to control for any word-level effects beyond those of distance (e.g., pragmatic factors), in a similar manner as for *task*.

 e. *Task*:*Word*, to control for the interaction between *task* and *word*, and particularly cases where a certain task is more likely to prompt the use of a certain word.

We tried adding other random effects, but this led to convergence issues, and even in cases where the models converged, their key results were the same as they were for these models. For more information, see Appendix S5 in S1 File (under "Models with alternative random effects").

Before building the models, we scaled the distance predictor by a factor of 10, so that it is on a scale of 0–10 instead of 0–1. This facilitates convergence, by putting this predictor on a similar scale as the other predictors (L2 proficiency: 1–12, frequency: ~1–7.5). We also centered the predictors, to facilitate convergence of the models and reduce potential collinearity.

After building the models, we exponentiated the coefficient estimates to derive an *incidence rate ratio* (IRR), and scaled the *standard errors* (SEs) accordingly [42]. The IRR is the expected change in the rate of the response as a factor of a 1-unit increase in the predictor. Accordingly, an IRR of 2 means a 1-unit increase in the predictor doubles the rate of use of the target word, while an IRR of 0.5 means a 1-unit increase in the predictor halves it. An IRR of 1 corresponds to a coefficient estimate (*B*) of 0. For more information, see "Incidence rate ratio" in Appendix S4 in S1 File.

In addition, we checked the statistical assumptions of the models. The relevant diagnostics appear in Appendix S4 in S1 File (under "Model diagnostics"), and indicate that there are no substantial issues with the models.

Finally, we also compared these models with baseline models, which did not include lexical distance as a predictor, to determine whether the inclusion of lexical distance improves the models' predictive power (based on AIC and BIC).

## Results

Fig 3 contains plots showing the basic association between distance and the rate of use of words in the datasets, compared to their baseline frequency in English. For the associated statistics, see "Frequency-ratio descriptive statistics" in Appendix S3 in S1 File.

If there is facilitative influence of crosslinguistic similarity, then we would expect words with a lower lexical distance (i.e., higher similarity) to have a higher frequency ratio. However, such effect is not visible in the plots, as the frequency ratio seems independent of lexical distance. Nevertheless, since this analysis is limited (e.g., it does not control for task effects), we move on to the more comprehensive mixed-models.

Table 5 contains the results of the mixed-models for the Swadesh lists. There is essentially no effect of distance or of its interaction with L2 proficiency, as the associated effect sizes are almost exactly zero (B = -0.01–0.00, corresponding to IRR = 0.99–1.00). Given this, and given that the associated SEs are also very small (≤0.01 for both B and IRR), this lack of effect is robust within this sample.

In addition, there is almost no variance between the L1s based on the associated random effect (SD ≤ 0.03), which suggests that speakers of different L1s used the target words in similar rates. However, this should be interpreted with caution, since this variance is likely underestimated due to the small number of L1s. Nevertheless, its exact magnitude is not crucial to our study, since we focus on the effects of distance, and as shown in Appendix S5 in S1 File (under "Models with alternative random effects"), the models' estimates remain functionally identical when the L1 random effect is not included.

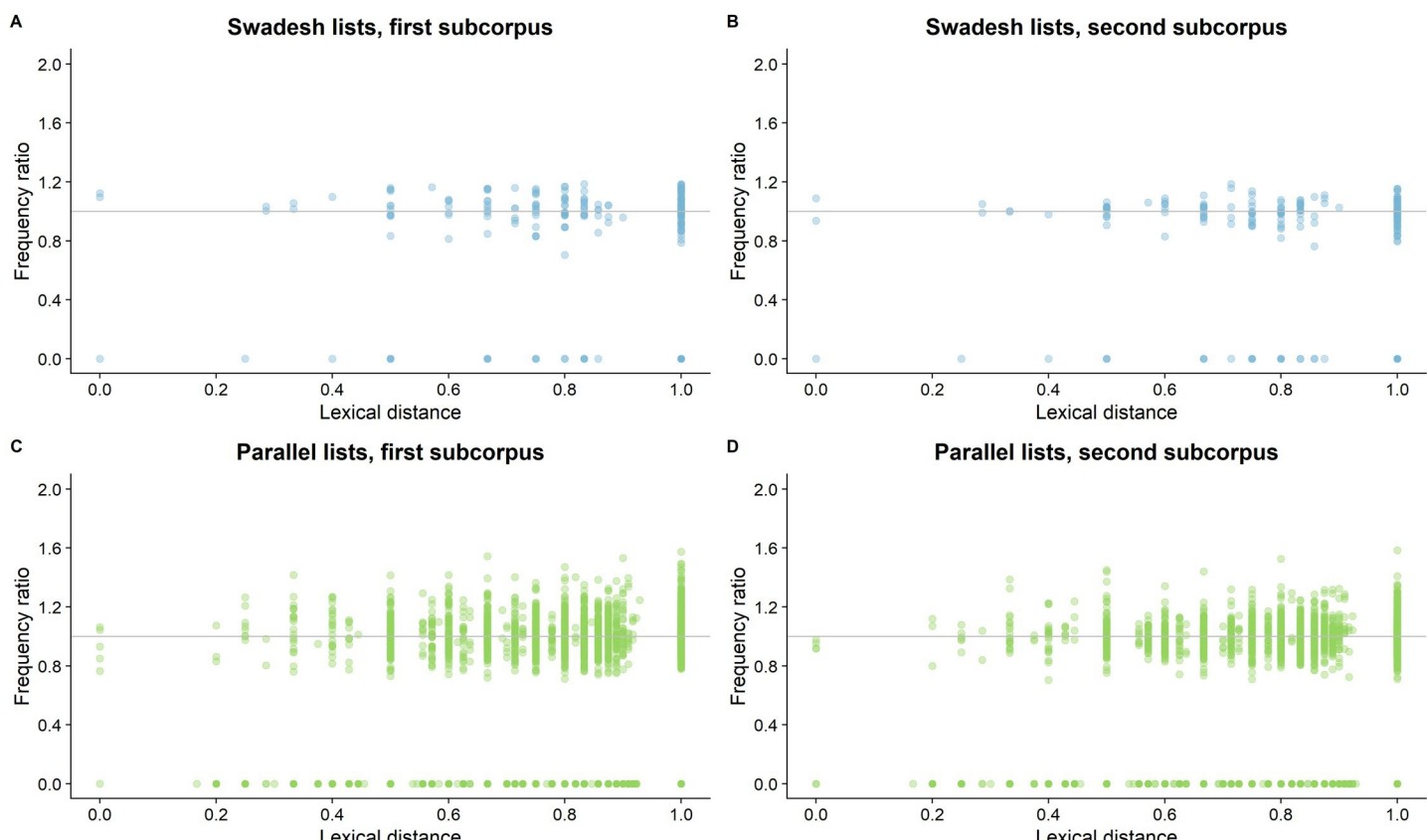

**Fig 3. The *lexical distance* of words and their *frequency ratio* (i.e., their frequency in the sample divided by their baseline frequency in English).** A ratio = 1 (grey line) indicates that a word is used in equal rates in our learner sample and baseline English; a ratio >1 indicates a word is used more frequently in our sample, and a ratio <1 indicates the opposite; a ratio = 0 indicates the word does not appear in our sample. Each point is a combination of a target word and a specific L1, since words in different L1s can have different distances from English. Darker shading indicates an overlap in points.

By contrast, the random effects of *task* and *word* are stronger than the *L1* effect by an order of magnitude or more (SD = 0.33–0.46), and the *task:word* effect is even stronger (SD = 1.36–1.84), which shows that these factors, and primarily the need to use specific words in specific tasks, have a much stronger influence on learners' rate of use of L2 words. Similarly, *frequency* as a control variable also has a very strong effect (B = 3.16–3.30, corresponding to IRR = 23.53–26.99), which was expected since the response variable is a type of frequency measure.

Table 6 contains the results of the mixed-models based on the parallel dictionaries. The findings of these models support those of the Swadesh-based models. Specifically, there is essentially no effect of distance or of its interaction with proficiency (B = 0.00–0.01, corresponding to IRR = 1.00–1.01), and the associated SEs are also very small ($\leq$0.01 for both B and IRR). In addition, as in the Swadesh-based models, there is almost no variance based on the *L1* random effect (SD $\leq$ 0.01), though the number of L1s included is even smaller, which again necessitates caution in the interpretation of the exact magnitude of this effect.

A minor difference is that there is lower variance in the *task* random effect here (SD = 0.03–0.11). However, there is also greater variance based on the *word* and *task:word* effects (SD = 0.45–0.65 and SD = 1.50–2.30 respectively). This supports the overall findings in this regard from the Swadesh models, which is that the need to use specific L2 words in specific

**Table 5. Results of the mixed-models, for the Swadesh-based samples.** The response variable was the rate of use of the target L2 English words (i.e., their count offset by the total number of words in each text). Under *fixed effects*, *distance* is the phonological LDN between each L2 word and its most lexically similar L1 counterpart (originally 0–1, scaled to 0–10), *proficiency* is the EFCAMDAT L2 proficiency level at which the text was written (1–12, corresponding to CEFR A1–B2), and *frequency* is the baseline Zipf frequency of the target word in English (~1–7.5). Under random effects, $\tau_{00}$ and $\tau_{11}$ respectively represent the SD of the associated random intercepts and slopes, and $\rho_{01}$ represents the correlation between random intercepts and associated random slopes (here, *distance* for *L1*).

| Predictor | First subcorpus | | | | | | Second subcorpus | | | | | |
|---|---|---|---|---|---|---|---|---|---|---|---|---|
| | $B$ | $SE_B$ | $IRR$ | $SE_{IRR}$ | $Z$ | $p$ | $B$ | $SE_B$ | $IRR$ | $SE_{IRR}$ | $z$ | $p$ |
| (Intercept) | -10.32 | 0.16 | 0.00 | <0.01 | -65.40 | < .001 | -9.86 | 0.14 | 0.00 | <0.01 | -68.45 | < .001 |
| Distance | -0.01 | 0.01 | 0.99 | 0.01 | -1.17 | .243 | -0.01 | 0.01 | 0.99 | 0.01 | -0.36 | .718 |
| Proficiency | -0.04 | 0.02 | 0.96 | 0.02 | -2.12 | .034 | 0.00 | 0.02 | 1.00 | 0.02 | -0.22 | .829 |
| Frequency | 3.30 | 0.21 | 26.99 | 5.66 | 15.70 | < .001 | 3.16 | 0.19 | 23.53 | 4.50 | 16.50 | < .001 |
| Dist:Prof | 0.00 | <0.01 | 1.00 | <0.01 | 0.61 | .543 | 0.00 | <0.01 | 1.00 | <0.01 | -1.28 | .202 |
| *Random effects* | | | | | | | | | | | | |
| Learner_$\tau_{00}$ | 0.07 | | | | | | 0.23 | | | | | |
| Task_$\tau_{00}$ | 0.40 | | | | | | 0.33 | | | | | |
| Word_$\tau_{00}$ | 0.38 | | | | | | 0.46 | | | | | |
| Task:Word_$\tau_{00}$ | 1.84 | | | | | | 1.36 | | | | | |
| L1_$\tau_{00}$ | 0.02 | | | | | | 0.03 | | | | | |
| L1.Distance_$\tau_{11}$ | 0.01 | | | | | | 0.03 | | | | | |
| L1_$\rho_{01}$ | 0.55 | | | | | | -0.14 | | | | | |

tasks strongly influences learners' tendency to use those words. Finally, and as expected, frequency is a substantial predictor here too (B = 2.89–2.97, IRR = 18.08–19.50).

The results of the models are summarized in Fig 4, which contains the fixed effects from each model, and which illustrates the lack of effect of lexical distance and of its interaction with L2 proficiency. Furthermore, these results are supported by the comparisons with the baseline models (with no lexical distance), which appear in Appendix S5 in S1 File (under "Baseline models").

**Table 6. Results of the mixed-models, for the parallel-based samples.** The response variable was the rate of use of the target L2 English words (i.e., their count offset by the total number of words in each text). Under *fixed effects*, *distance* is the phonological LDN between each L2 word and its most lexically similar L1 counterpart (originally 0–1, scaled to 0–10), *proficiency* is the EFCAMDAT L2 proficiency level at which the text was written (1–12, corresponding to CEFR A1–B2), and *frequency* is the baseline Zipf frequency of the target word in English (~1–7.5). Under random effects, $\tau_{00}$ and $\tau_{11}$ respectively represent the SD of the associated random intercepts and slopes, and $\rho_{01}$ represents the correlation between random intercepts and associated random slopes (here, *distance* for *L1*).

| Predictor | First subcorpus | | | | | | Second subcorpus | | | | | |
|---|---|---|---|---|---|---|---|---|---|---|---|---|
| | $B$ | $SE_B$ | $IRR$ | $SE_{IRR}$ | $Z$ | $p$ | $B$ | $SE_B$ | $IRR$ | $SE_{IRR}$ | $z$ | $p$ |
| (Intercept) | -12.85 | 0.06 | 0.00 | <0.01 | -207.79 | < .001 | -12.59 | 0.05 | 0.00 | <0.01 | -243.41 | < .001 |
| Distance | 0.01 | <0.01 | 1.01 | <0.01 | 1.91 | .056 | 0.01 | 0.01 | 1.01 | 0.01 | 1.04 | .301 |
| Proficiency | 0.11 | 0.01 | 1.12 | 0.01 | 9.22 | < .001 | 0.04 | 0.01 | 1.04 | 0.01 | 4.29 | < .001 |
| Frequency | 2.89 | 0.06 | 18.08 | 1.05 | 49.86 | < .001 | 2.97 | 0.05 | 19.50 | 0.99 | 58.52 | < .001 |
| Dist:Prof | 0.00 | <0.01 | 1.00 | <0.01 | 1.25 | .211 | 0.00 | <0.01 | 1.00 | <0.01 | 1.09 | .276 |
| *Random effects* | | | | | | | | | | | | |
| Learner_$\tau_{00}$ | 0.03 | | | | | | 0.04 | | | | | |
| Task_$\tau_{00}$ | 0.03 | | | | | | 0.11 | | | | | |
| Word_$\tau_{00}$ | 0.45 | | | | | | 0.65 | | | | | |
| Task:Word_$\tau_{00}$ | 2.30 | | | | | | 1.50 | | | | | |
| L1_$\tau_{00}$ | 0.00 | | | | | | 0.01 | | | | | |
| L1.Distance_$\tau_{11}$ | 0.01 | | | | | | 0.01 | | | | | |
| L1_$\rho_{01}$ | 0.25 | | | | | | 0.81 | | | | | |

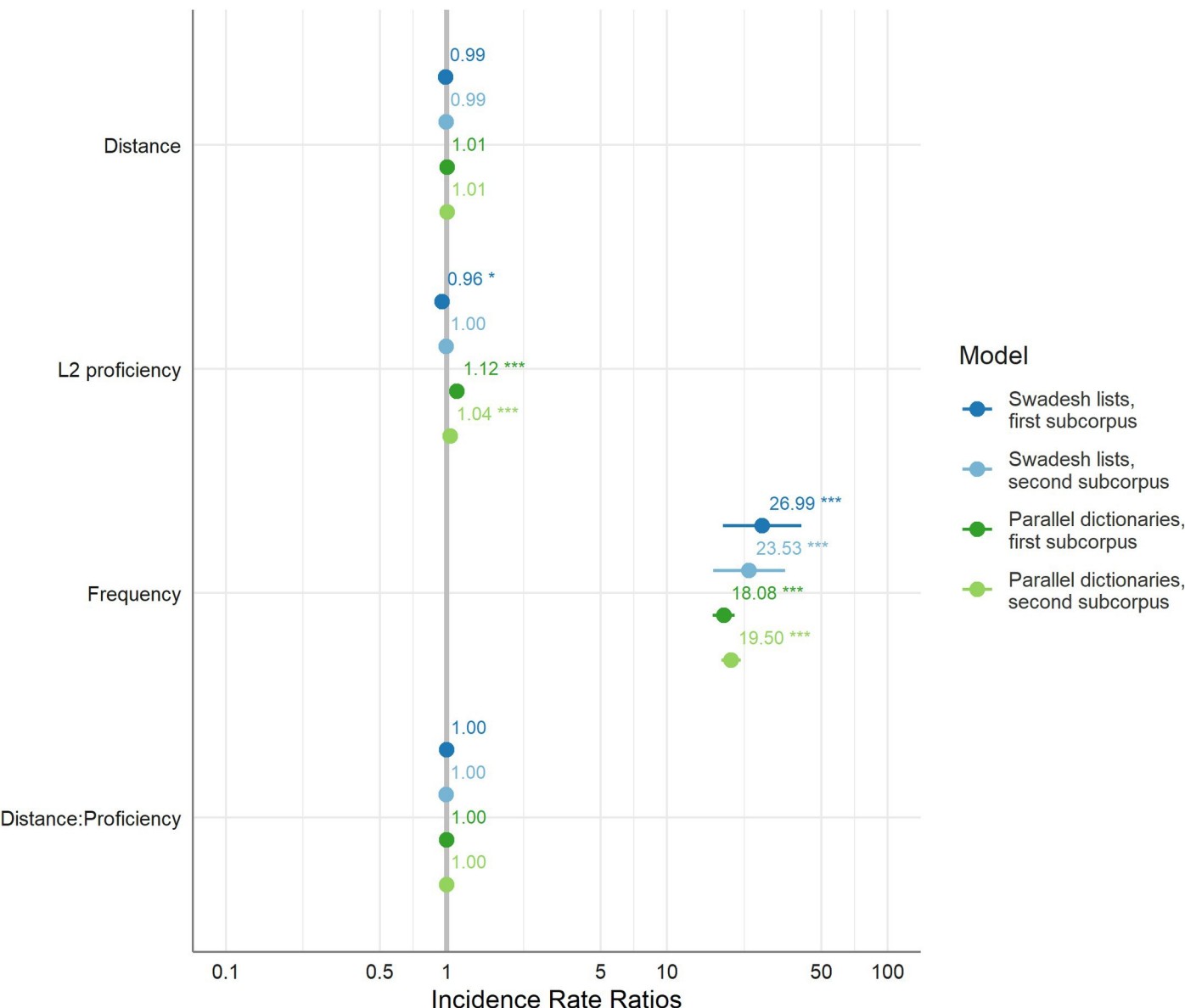

**Fig 4. The models' fixed effects, illustrating the lack of effect of lexical distance and its interaction with L2 proficiency.** *Distance* is the phonological LDN between each L2 word and its most lexically similar L1 counterpart (scaled to 0–10), *proficiency* is the EFCAMDAT L2 proficiency level at which the text was written (scale of 1–12), and *frequency* is the baseline Zipf frequency of the target word in English (scale of ~1–7.5). Dots denote the IRR. Lines denote the 95% CIs; where they seem missing, it is because they are very narrow. Asterisks denote statistical significance of the coefficient estimate (* denotes $p < .05$ and *** denotes $p < .001$).

Note that the effects of proficiency itself were weak and inconsistent across the models. However, it appears that there is a weak positive effect of proficiency for the parallel-dictionaries samples, likely because they contain some words that are lower-frequency than in the Swadesh lists. This suggests there is an interaction between proficiency and frequency, and this is supported by the "Added-interactions models" shown in Appendix S5 in S1 File, which also show that including this interaction in the models causes convergence issues, and does not change our key findings.

## Discussion

### Study summary

We investigated whether formal crosslinguistic lexical similarity (phonological overlap) between L1 words and their L2 counterparts increases the use of the L2 words in a task-based educational setting, and whether this is moderated by L2 proficiency.

We found no effect of crosslinguistic similarity on L2 vocabulary use, and no interaction between lexical similarity and L2 proficiency. This null finding was robust across all the combinations of the two subcorpora and two lexical-distance datasets that we examined, since all the associated predictors were tightly clustered around an IRR of 1 (corresponding to a coefficient estimate of 0), as shown in Tables 5 and 6 and in Fig 4. In addition, there was very low variance between the L1s based on the associated random effect (Tables 5 and 6), which suggests that speakers of different L1 used the target words in similar rates, despite the variation in the average lexical distance between them (shown in Fig 1 and Table 1). Conversely, the *task*, *word*, and especially the *task:word* random effects strongly influenced learner's word choices (Tables 5 and 6), which shows that these factors, and primarily the need to use specific words in specific tasks, have a much stronger influence on people's L2 vocabulary choices.

In addition, these results replicated across a range of supplementary analyses that we conducted, which appear in Appendices S2 and S5 in S1 File. These include models using feature edit distance, models using German-only data, models using a binary response variable, and models with added interactions.

### Main implications

The main implication of our findings is that formal crosslinguistic lexical similarity (in this case, phonological overlap), which relates to cognancy, does not influence learners' L2 productions in the type of constrained task-based educational setting we examined, which many L2 learners are likely to encounter. This is regardless of learners' L2 proficiency, and applies to learners at the A1–B2 CEFR range of L2 proficiency, though the complete lack of interaction between lexical similarity and L2 proficiency that we found suggests that this likely applies also to learners at the C1–C2 range of proficiency.

This finding supports the lexical intergroup homogeneity that Crossley and McNamara found among speakers of different L1s in a task-based setting (the ICLE) [25]. This suggests that the lack of L1 effect that they found is not due to their use of a global lexical measure (lexical diversity) or an idiosyncrasy in their sample, but is rather more likely a general feature of L2 lexical production in constrained task-based settings.

At the same time, this does not necessarily contradict studies that found an L1 effect on L2 word choice independently of crosslinguistic lexical similarity (e.g., in stylometry). Rather, the difference may be that the L1 effect found in those studies was driven by factors other than crosslinguistic similarity, such as a strong cultural preference for certain words (e.g., *hockey*), or that there were weaker task effects in their samples (e.g., because the prompts were less constrained).

Our finding also does not necessarily contradict the studies that found an effect of lexical similarity on the processing of individual L2 words or on broad L2 acquisition. Rather, it shows that this effect is different in this specific form of L2 production, where word choice is primarily driven by task-related factors, such as a specific message the learner needs to communicate. This interpretation is supported by the strong effects of *task*, *word*, and *task:word* on word choice, which suggest that the need to use a specific word for a specific task is what drives

learners' decision of whether to use it in the present context, regardless of whether the word is similar to their L1.

Accordingly, although L2 words that are similar to their L1 translation are likely easier for learners to access and use, the communicative needs of tasks can override this crosslinguistic influence, and drive learners to use necessary words rather than easier ones. This means that even if the facilitative effect of L1 similarity is there, which we expect is the case, its influence is too weak to drive learners' word choice in the present setting.

In addition, it is likely that other aspects of the tasks and their educational context played a role in determining word choice, and can play a role in similar contexts (especially—but not only—educational ones). For example, it is likely that the lessons associated with tasks involved words (i.e., *content*) that learners then used for practice, or that some task prompts elicited the use of a specific register (i.e., *style*) that necessitated the use of certain words. This supports and extends limited past research which found that factors such as formality and task type may influence transfer [21], and highlights the importance of considering these situational and contextual factors when investigating transfer.

Finally, note that past studies on the EFCAMDAT found L1 transfer effects on various other linguistic structures and phenomena, including clause subordination [46], relative clauses [47], clause-initial prepositional phrases [48], grammatical morphemes [49], articles [50], and capitalization [44]. X. Jiang et al. even found evidence of lexical transfer on the usage rates of certain punctuation marks (e.g., dashes) and phrases (e.g., "to my mind") [48].

The reason why they found an effect in this sample whereas we did not could be that the types of transfer involved in the structures they examined might not be as strongly influenced by communicative needs and task effects. For example, if a speaker wants to convey the meaning "I ate an apple", saying "apple" (a key content word) is generally more important than saying "an" (a functional element), since "I ate apple" conveys the original meaning more clearly than "I ate an". Alternatively, another potential—and not mutually exclusive—explanation for the difference in the finding is that negative transfer (which was the focus of most of those past studies) may be "stronger" from a cognitive perspective than positive transfer (which was the focus of the present study), and therefore more difficult for communicative needs and task effects to override. This ties in to earlier discussions on the differences between these types of transfer [51].

## Task effects in lexical choices

The strong task effects that were found in this study contribute to the growing evidence on the role of these effects in L2 lexical choices [39, 45, 52–55]. This highlights the importance of controlling for such effects (e.g., the purpose or context of production) when analyzing L2 lexical choices, particularly in learner corpora, where they can often play a substantial role.

## Limitations and future research

One limitation of this study is the use of one learner sample, so the analyses should be replicated on other samples, to determine the generalizability of the findings. Such replications can, for example, analyze speaking (rather than writing), analyze a different L2 (since English is a lingua franca), or analyze productions in other settings. It will be particularly beneficial to analyze L2 productions from learners who are writing in similar general settings, but under different levels of the communicative-constraints spectrum. Likewise, it would be interesting to compare written L2 productions to spoken ones, when these are produced by similar learners under similar conditions. This will show whether and how the effect of this crosslinguistic similarity, if it appears, varies across these two modes of language production.

Other limitations are the use of LDN, which does not directly capture information such as cognancy status, and the use of L2 words that often did not appear in learners' writing. Given all the information we presented (e.g., regarding the distribution of the response variable), we do not think that these limitations explain the null effect that we found. Nevertheless, it will be beneficial to replicate our analyses using other lexical-distance datasets and measures. It will be particularly beneficial to use a dataset such as *CogNet*, to examine the effects of cognancy directly, and to analyze more L2 words.

When doing this, it is also possible to focus on preference for cognates within sets of synonyms corresponding to the same meaning, similarly to Rabinovich et al. [13]. As discussed in Appendix S3 in S1 File (under "Analysis of synonym sets"), this can be done by comparing, within each set, the probability that speakers of different L1s will use any given synonym, and checking if their choices reflect a preference for cognates.

In addition, future research could also refine these analyses by accounting for further factors. For example, it might be beneficial to look at the baseline L1 frequency of words within specific genres that correspond to the associated writing tasks, rather than in the L1 as a whole. Similarly, it may be beneficial to examine the effects of genre and formality on the crosslinguistic influence that learners display in their L2 productions.

Finally, future research could also address the questions outlined in the discussion of the study's main implications. Notably, this could involve comparing the effects of communicative needs on different types of transfer, such as positive vs. negative transfer, or lexical vs. syntactic transfer.

## Conclusions

In the present task-based educational settings, formal lexical similarity—which relates to cognancy and which we based on phonological overlap between corresponding L1-L2 words—did not influence L2 word choice, regardless of learners' L2 proficiency. This suggests that the effects of formal lexical similarity are more constrained than expected, and that communicative needs and task effects can sometimes override the influence of positive lexical transfer. This raises questions regarding when and how communicative needs and task effects influence language transfer, for example in different types of transfer (e.g., lexical vs. syntactic, positive vs. negative).

## Supporting information

**S1 File. Supplementary appendices.** This includes Appendix S1 (Lexical distance), Appendix S2 (Feature edit distance), Appendix S3 (Additional descriptive information), Appendix S4 (Additional technical information), and Appendix S5 (Additional models).
(PDF)

## Acknowledgments

This research is based on the PhD thesis of the first author [56].

## Author Contributions

**Conceptualization:** Itamar Shatz.

**Data curation:** Itamar Shatz.

**Formal analysis:** Itamar Shatz.

**Investigation:** Itamar Shatz.

**Methodology:** Itamar Shatz, Theodora Alexopoulou, Akira Murakami.

**Visualization:** Itamar Shatz.

**Writing – original draft:** Itamar Shatz.

**Writing – review & editing:** Itamar Shatz, Theodora Alexopoulou, Akira Murakami.

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
