## [Decision Letter · Decision Letter 0]

8 Jan 2023

PONE-D-22-31186The Potential Influence of Crosslinguistic Similarity on Lexical Transfer: Examining Vocabulary Choice in L2 EnglishPLOS ONE

Dear Dr. Shatz,

Thank you for submitting your manuscript to PLOS ONE. After careful consideration, we feel that it has merit but does not fully meet PLOS ONE’s publication criteria as it currently stands. Therefore, we invite you to submit a revised version of the manuscript that addresses the points raised during the review process.

We look forward to receiving your revised manuscript.

Kind regards,

Ramona Bongelli, Ph.D.

Academic Editor

PLOS ONE

Journal Requirements:

2. We noted in your submission details that a portion of your manuscript may have been presented or published elsewhere. [Yes; the full content of the paper (with some modifications) has been published as part of the first author's PhD thesis (see Ch. 4), available at https://www.repository.cam.ac.uk/handle/1810/339032

This does not constitute dual publication based on PLOS's guidelines (https://journals.plos.org/plosone/s/ethical-publishing-practice), which state that "Prior publication of research as a thesis, presentation at medical or scientific conferences, or posting on preprint servers will not preclude consideration of your manuscript."] Please clarify whether this publication was peer-reviewed and formally published. If this work was previously peer-reviewed and published, in the cover letter please provide the reason that this work does not constitute dual publication and should be included in the current manuscript.

Reviewers' comments:

Reviewer's Responses to Questions

**Comments to the Author**

1. Is the manuscript technically sound, and do the data support the conclusions?

Reviewer #1: Yes

Reviewer #2: Yes

2. Has the statistical analysis been performed appropriately and rigorously? 

Reviewer #1: Yes

Reviewer #2: Yes

3. Have the authors made all data underlying the findings in their manuscript fully available?

Reviewer #1: Yes

Reviewer #2: Yes

4. Is the manuscript presented in an intelligible fashion and written in standard English?

Reviewer #1: Yes

Reviewer #2: Yes

5. Review Comments to the Author

Reviewer #1: Summary:

Using EFL learners corpora, the authors examine the effect of cross-linguistic similarity on the production of L2 words. The authors further examine if this relationship is influenced by the learners’ L2 proficiency. The results show that lexical distance, language background (i.e., the learners’ L1) and L2 proficiency do not influence word choices. Rather, word choice seems to be determined by task.

Overall, I found the paper very clearly written (especially the methods and results sections), and I enjoyed reading it. I also really appreciate the Open Science practices adopted by the authors as well as how they clearly integrate them in the body of the manuscript. I only have a few minor comments.

Overall comments:

Abstract:

-I find it difficult to understand what modelling the similarity on the rate of use of L2 words means. Could the authors reformulate?

Introduction:

-I was wondering if the authors could define more clearly similarity in the introduction, and particularly focus on the type of similarity they focus on in the paper. For now, the introduction is rather focused on cognates.

-Could the authors define cognancy upon first mention? (and in the abstract as well if possible)

-Could the authors elaborate a bit on why lexical transfer can affect language use and acquisition either positively or negatively?

-I also think it would be useful to add a paragraph on how L2 proficiency could influence the relationship between cross-linguistic similarity and L2 word choices based on previous research, given that this is one of the two main research questions.

Analysis:

-What did the authors use to perform the model comparisons (p.23)?

Discussion:

-Could it be that the authors observed no effect of lexical distance because the learners completed the task in the written modality? Would the authors expect differences between written and spoken production? Especially given that similarity is partly calculated based on phonological information. It could also be that writing relies on more conscious processes, whereby learners are more likely to (have time to) think about which word to use to meet the task requirement). In contrast, learners could be more influenced could be influenced by less conscious processes during (faster) spoken production, and thus be more influenced by factors such as phonological overlap during spoken production.

Minor comments:

-“the goal and context of communication generally play a greater role in L2 production (e.g., word choice in essays) than in many experimental processing paradigms (e.g., reaction time to isolated words)” (p.3): can the authors include references to support these claims?

-“There is evidence that increased lexical similarity between languages—as measured through the mean similarity of L1-L2 word pairs”�what type of similarity are the authors referring to here?

-footnote 12: what do the authors mean with “although this does not substantially influence our findings” (p.22)

-I spotted a few typos:

-p.6: “is a simply feature of”

-p.18: “show that the models do are”

-p.30: “analsyes”

Reviewer #2: This is an interesting paper which sought to establish whether L1-L2 formal lexical similarity affects L2 word choice. Taking phonological overlap between L1 words and their English translations as the point of departure, the authors modelled the influence of similarity on the use of L2 words, using mixed-effects statistical methods. The authors explain in depth the concepts and methods applied in the study, and the presentation of the data is clear and comprehensible. The findings enrich the existing body of research on crosslinguistic influence, in particular that focusing on lexical transfer, and point to the need to control for task effects in future studies.

I recommend the paper for publication.

Below is a list of editorial inaccuracies which require the authors’ attention:

p. 6

is a simply feature

p. 16

favorite [American spelling]

p. 18

appear in in Table 4

the count-based models [missing full stop]

show that the models do are

The authors may also want to consider the following observations:

The authors state that they consider only one aspect of formal similarity (phonological overlap), while disregarding other factors which may also affect lexical transfer (orthographic depth, semantic/pragmatic similarity). Given that the study examines written material and not oral production, wouldn't orthographic depth, rather than phonological overlap, be more relevant?

Word frequency

The authors may want to consider the potential interaction between frequency and genre/ text type.

The baseline frequency relied on in the study does not consider the in/frequency of individual lexical items in the genres/text types that the learners were required to produce.

The authors note the task:word effect and conclude that the need to use specific words in specific tasks has a strong influence on learners’ rate of use of L2 words. They also reference earlier research (Jarvis & Pavlenko 2008) which found that formality and task type may influence transfer. It is therefore recommended that they control for the effects of the purpose and context of production, i.e. mode of communication (spoken/written), genre (text type) and register (degree of formality) in future studies.

6. PLOS authors have the option to publish the peer review history of their article (what does this mean?). If published, this will include your full peer review and any attached files.

Reviewer #1: No

Reviewer #2: No

---

## [Author Response · Author response to Decision Letter 0]

12 Jan 2023

We would like to thank the editor and reviewers for their time.

Below, we outline how we revised the manuscript to address the reviewers’ comments. Note that:

> This mark indicates a reviewer’s comment.

— This mark indicates our response.

Reviewer #1

> Abstract:

> I find it difficult to understand what modelling the similarity on the rate of use of L2 words means. Could the authors reformulate?

— Reformulated it to read as follows: “We then used mixed-effects statistical models to examine how this similarity influences the rate of use of the L2 words; essentially, we checked whether L2 words that are more similar to their L1 translations are used more often.” We also added the following line right after this: “We also controlled for potential confounds, including the baseline L1 frequency of the English words.”

> Introduction:

> I was wondering if the authors could define more clearly similarity in the introduction, and particularly focus on the type of similarity they focus on in the paper. For now, the introduction is rather focused on cognates.

— Added more of an explanation about this: ‘This similarity is usually conceptualized based on the overlap in sounds and/or letters between words in different languages. For example, the French word for “orange” is also spelled “orange” (though pronounced slightly differently), so it has higher formal similarity with its English translation than does the French word for “lemon” (“citron”). Accordingly, it will generally be easier for French speakers to acquire the English word “orange” than the word “lemon”.’

> Could the authors define cognancy upon first mention? (and in the abstract as well if possible)

— We updated the introduction and abstract as suggested:

- Introduction: “When two words with similar meanings across languages have a high level of formal similarity, they can be considered to be psycholinguistic cognates, though there is no exact threshold for cognancy based on similarity. Psycholinguistic cognancy frequently occurs because the words are also historical cognates, meaning that they share a common etymology, though cognancy may also involve words borrowed through language contact [11,13,14].”

- Abstract: “We quantified similarity based on phonological overlap between L1 words and their L2 (English) translations. This similarity relates to psycholinguistic cognancy, which occurs when words and their translations share a high level of formal similarity, often due to historical cognancy from shared etymology or language contact.

> Could the authors elaborate a bit on why lexical transfer can affect language use and acquisition either positively or negatively?

— Updated the introduction to include this, so it now reads as follows: “This is often attributed to lexical transfer [1], a type of language transfer or crosslinguistic influence [2–4]. Transfer can be positive when it facilitates language acquisition or use, for example because an L2 linguistic structure (e.g., a certain word) is identical to a corresponding structure in a learner’s L1, which makes it easy for the learner to remember it. Transfer can also be negative when it hinders language acquisition or use, in which case it is also called interference; this can occur, for example, because an L2 structure is very different from a corresponding structure in a learner’s L1, which makes it hard for the learner to remember it.”

> I also think it would be useful to add a paragraph on how L2 proficiency could influence the relationship between cross-linguistic similarity and L2 word choices based on previous research, given that this is one of the two main research questions.

— We now have the following paragraph in the introduction: “Like most types of crosslinguistic influence, this form of lexical transfer is expected to play a role primarily during early stages of SLA, when learners rely more on their L1 to form and use their mental lexicon. However, this influence can also play a role at advanced stages of SLA, and therefore affect even advanced L2 learners [1,5,6,10,15].”

> Analysis:

> What did the authors use to perform the model comparisons (p.23)?

— Added a clarification that this is “based on AIC and BIC” (we elaborate on these measures in §5.2 of the SI, where we present the comparisons and provide relevant references).

> Discussion:

> Could it be that the authors observed no effect of lexical distance because the learners completed the task in the written modality? Would the authors expect differences between written and spoken production? Especially given that similarity is partly calculated based on phonological information. It could also be that writing relies on more conscious processes, whereby learners are more likely to (have time to) think about which word to use to meet the task requirement). In contrast, learners could be more influenced could be influenced by less conscious processes during (faster) spoken production, and thus be more influenced by factors such as phonological overlap during spoken production.

— We agree that the fact that learners were writing, rather than speaking, may have affected their productions in a way that could have weakened any potential effect of crosslinguistic similarity for the reasons you described. However, we don’t think that this alone can explain the complete lack of effect in the study, given that past research on this effect did find it in written productions (especially the research on word-choice transfer which is the most similar to ours, like Jarvis et al., 2012 and Rabinovich et al., 2018). Nevertheless, since this is an interesting an important question to address, we added the following to our “Limitations and future research” section: “Likewise, it would be interesting to compare written L2 productions to spoken ones, when these are produced by similar learners under similar conditions. This will show whether and how the effect of this crosslinguistic similarity, if it appears, varies across these two modes of language production.” (We focused on the future research, rather than the potential mechanism, to avoid being too speculative. Also, note that this was previously mentioned more briefly at the end of this section, so we consolidated the previous mention with the new one.)

— In addition, we now note in the body of the paper (under “Calculating lexical distance”) that there is a strong correlation between phonological and orthographic overlap in our sample (r = .68, 95% CI = [.67, .70], p < .001; this is for the parallel dictionaries, where all the L1s share English’s Latin script), and that a similar correlation was found in other studies (e.g., r = .782 in Carrasco-Ortiz et al., 2021). Accordingly, the phonological distances that we used were also highly indicative of the orthographic distances between words. (This was previously included only in a footnote in the SI, but given the comments by you and the other reviewer, we added a mention of this to the paper itself.) This further suggests that the lack of effect here is not solely due to the productions being written, especially given how robust the null effect was. 

> Minor comments:

> “the goal and context of communication generally play a greater role in L2 production (e.g., word choice in essays) than in many experimental processing paradigms (e.g., reaction time to isolated words)” (p.3): can the authors include references to support these claims?

— This is a general observation that we made based on our experience, but since we don’t have specific references to support it, we moderated the claim to show its speculate nature, so we now say “might play a greater role” rather than “generally play a greater role”. 

> “There is evidence that increased lexical similarity between languages—as measured through the mean similarity of L1-L2 word pairs” what type of similarity are the authors referring to here?

— The studies that we cite there used several different measures. Since this isn’t the focus of the statement, and we don’t want to get sidetracked by explaining the different measures there, we simplified this to: “There is evidence that increased overall lexical similarity between languages improves learning outcomes, thus leading to higher scores in L2 proficiency tests [17–19].”

> footnote 12: what do the authors mean with “although this does not substantially influence our findings” (p.22)

— Edited the material so it should be clearer: ‘We tried adding other random effects, but this led to convergence issues, and even in cases where the models converged, their key results were the same as they were for these models. For more information, see Appendix S5 (under “Models with alternative random effects”).’ (Note that this is now mentioned in the body, rather than in a footnote, in line with PLOS’s style requirements).

> I spotted a few typos:

> p.6: “is a simply feature of”

> p.18: “show that the models do are”

> p.30: “analsyes”

— Fixed these.

Reviewer #2

> Editorial inaccuracies which require the authors’ attention:

> p. 6, is a simply feature

> p. 16, favorite [American spelling]

> p. 18, appear in in Table 4; the count-based models [missing full stop]; show that the models do are

— Fixed these.

> The authors may also want to consider the following observations:

> The authors state that they consider only one aspect of formal similarity (phonological overlap), while disregarding other factors which may also affect lexical transfer (orthographic depth, semantic/pragmatic similarity). Given that the study examines written material and not oral production, wouldn't orthographic depth, rather than phonological overlap, be more relevant?

— We added the following explanation to the body of the paper to address this (under “Calculating lexical distance”): “We used phonological—rather than orthographic—overlap as a measure of distance, because this enables us to examine distance from L1s that have a substantially different script from English, like Arabic and Mandarin. Nevertheless, we also examined a sample (presented later under “Learner sample”) where all the L1s share English’s Latin script, and in this sample, there was a strong correlation between phonological and orthographic overlap (r = .68, 95% CI = [.67, .70], p < .001). This aligns with findings of other research [11,29,30], like an r = .782 found in a dataset of English and Spanish words [11]. This strong correlation suggests that the phonological overlap that we found for in L1s that share English’s script is indicative of the associated orthographic overlap in these L1s, so even if a large part of the effect of similarity is due to overlap in orthography, we would expect to detect it in our analyses.” (Note: Some of this material was/is also included as a footnote in the SI, but given the comment we thought that it would help to highlight it in the body of the paper.)

> Word frequency

> The authors may want to consider the potential interaction between frequency and genre/ text type. The baseline frequency relied on in the study does not consider the in/frequency of individual lexical items in the genres/text types that the learners were required to produce.

— We added this to the “Limitations and future research” section (see our response to the next comment). We agree that this might be beneficial for refining the analyses further, but it currently isn’t feasible to do this in our sample, for two reasons. First, there is no existing classification that covers all the tasks (especially in the EFCAMDAT Cleaned Subcorpus), and creating one would necessitate a full extensive study of its own. Second, even once such a classification exists, it would be necessary to get corresponding texts produced by L1 speakers under similar conditions, which would necessitate another huge research project, especially given the number of tasks in this dataset. We could potentially add a random effect between task and frequency to sort of account for this instead, but based on all the alternative models that we built (see §5.1 in the SI), this will very likely cause convergence issues. This also has other downsides, like reducing the interpretability of this predictor, and making it harder to compare it to corresponding effects found in other studies. Nevertheless, given that the frequency measure that we used was a good predictor in our models, and given the robustness of our main findings, this is very unlikely to change our key findings.

> The authors note the task:word effect and conclude that the need to use specific words in specific tasks has a strong influence on learners’ rate of use of L2 words. They also reference earlier research (Jarvis & Pavlenko 2008) which found that formality and task type may influence transfer. It is therefore recommended that they control for the effects of the purpose and context of production, i.e. mode of communication (spoken/written), genre (text type) and register (degree of formality) in future studies.

— Agreed, so we added the following paragraph to the “Limitations and future research” section: “In addition, future research could also refine these analyses by accounting for further factors. For example, it might be beneficial to look at the baseline L1 frequency of words within specific genres that correspond to the associated writing tasks, rather than in the L1 as a whole. Similarly, it may be beneficial to examine the effects of the genre of text and its degree of formality on the crosslinguistic influence that learners display.” We also mention the point regarding written vs. spoken communication a bit earlier in this section, in response to a comment by the other reviewer: “Likewise, it would be interesting to compare written L2 productions to spoken ones, when these are produced by similar learners under similar conditions. This will show whether and how the effect of this crosslinguistic similarity, if it appears, varies across these two modes of language production.

---

## [Editor Report · Decision Letter 1]

16 Jan 2023

Examining the potential influence of crosslinguistic lexical similarity on word-choice transfer in L2 English

PONE-D-22-31186R1

Dear Dr. Shatz,

We’re pleased to inform you that your manuscript has been judged scientifically suitable for publication and will be formally accepted for publication once it meets all outstanding technical requirements.

Kind regards,

Ramona Bongelli, Ph.D.

Academic Editor

PLOS ONE
---

## [Editor Report · Acceptance letter]

23 Jan 2023

PONE-D-22-31186R1 

Examining the potential influence of crosslinguistic lexical similarity on word-choice transfer in L2 English 

Dear Dr. Shatz:

I'm pleased to inform you that your manuscript has been deemed suitable for publication in PLOS ONE. Congratulations! Your manuscript is now with our production department. 

Kind regards, 

on behalf of

Professor Ramona Bongelli 

Academic Editor

PLOS ONE